# In Vitro Metabolism of Helenalin Acetate and 11α,13-Dihydrohelenalin Acetate: Natural Sesquiterpene Lactones from Arnica

**DOI:** 10.3390/metabo12010088

**Published:** 2022-01-17

**Authors:** Franziska M. Jürgens, Matthias Behrens, Hans-Ulrich Humpf, Sara M. Robledo, Thomas J. Schmidt

**Affiliations:** 1Institute of Pharmaceutical Biology and Phytochemistry, University of Münster, Corrensstrasse 48, D-48149 Münster, Germany; franziska.juergens@uni-muenster.de; 2Institute of Food Chemistry, University of Münster, Corrensstraße 45, D-48149 Münster, Germany; mattbehrens@uni-muenster.de (M.B.); humpf@uni-muenster.de (H.-U.H.); 3PECET-School of Medicine, University of Antioquia, Calle 70 N° 52-21, Medellin 0500100, Colombia; sara.robledo@udea.edu.co

**Keywords:** in vitro metabolism, liver microsomes, glutathione conjugation, helenalin derivatives, sesquiterpene lactones, *Arnica montana*

## Abstract

Arnica tincture is a herbal medicinal preparation with anti-inflammatory activity which is used traditionally for the topical treatment of blunt injuries as well as rheumatic muscle and joint complaints. Its main bioactive constituents are sesquiterpene lactones (STLs) of the helenalin and 11α,13-dihydrohelenalin types. Besides the mentioned activity, the tincture and its isolated STLs have antileishmanial activity. In a recent in vivo study, a treatment with Arnica tincture cured cutaneous Leishmaniasis (CL) in a golden hamster model. CL is a neglected tropical disease affecting more than two million people every year, for which new treatments are urgently needed. In order to use Arnica tincture on open CL lesions of human patients, it is important to know how the constituents are metabolized. Therefore, in vitro metabolism experiments with liver microsomes of different species (rat, pig and human) were performed with the Arnica STLs helenalin acetate and 11α,13-dihydrohelenalin acetate. Phase I and phase II metabolism experiments were performed, as well as a combination of both. Glutathione conjugation plays a major role in the metabolism of these STLs, as could be expected based on previous reports on their reactivity. Besides glutathione conjugates, several other metabolites were formed, e.g., water conjugates and hydroxides. Our results show for the first time a detailed picture of the metabolism of Arnica STLs. The fast and extensive formation of glutathione conjugates makes it unlikely that low absorbed levels of these compounds, as expected after dermal absorption from Arnica tincture, could be of toxicological concern.

## 1. Introduction

Cutaneous Leishmaiasis (CL) is classified as 1 of 20 neglected tropical diseases by the World Health Organization (WHO). This vector-borne disease is endemic in several countries, especially in South America, North Africa and Southeast Asia, but also in Mediterranean countries [1,2]. Around 350 million people are at risk of a CL infection and every year 2–2.5 million new cases are reported [3]. CL is caused by protozoan parasites of different *Leishmania* (L.) species, which are transmitted by the bite of infected female phlebotomine sandflies [4]. After injection of the parasites into the skin, lesions arise that ulcerate during the progression of the disease. In severe cases with many lesions, patients suffer from serious disabilities and stigma due to life-long scars [2]. Standard treatments of CL are intralesional or parenteral injections of antimonial drugs, e.g., sodium stibogluconate and meglumine antimonate, that cause high systemic toxicity [5]. Consequently, new, efficacious, safe and affordable therapies to treat this neglected tropical disease are urgently needed. In in vitro studies, Arnica tincture and isolated Arnica sesquiterpene lactones (STLs) demonstrated strong antileishmanial activity [6,7]. In addition, a first in vivo study with CL in golden hamsters infected with *L. braziliensis* showed curative activity of an ethanolic Arnica tincture, applied topically on the lesions [8]. The efficacy was comparable to, and even somewhat better than, that of the standard drug meglumine antimonate after intralesional injection [8].

The traditional herbal medicinal product Arnica tincture, according to the European Pharmacopoeia (Ph. Eur.), is prepared as a 70% ethanolic extract from Arnica flowers Ph. Eur. (flowerheads of *Arnica montana* L., Asteraceae) [9]. It is used topically to treat blunt injuries and traumas, inflammations and rheumatic muscle and joint complaints [10]. Main bioactive substances are STLs with pseudoguaianolide structures. Among them, various esters of helenalin and 11α,13-dihydrohelenalin with low-molecular weight carboxylic acids, such as acetic, isobutyric, methacrylic, methylbutyric and tiglic acid, occur as main constituents besides small amounts of the unesterified parent STLs [8,10,11,12]. Apart from ethanol, dichloromethane is commonly used for STL extraction for isolation purposes and was also utilized in this work [13,14,15].

Reactive structural elements of the Arnica STLs are the Michael acceptor systems (α,β-unsaturated ketone moiety and α,β-unsaturated lactone moiety), the ester bond and the lactone moiety; hence, expected phase I reactions are, e.g., hydration, ester hydrolysis and oxidative modification, such as hydroxylation or epoxidation. As the structures contain reactive Michael acceptor systems, these compounds are known to react with free cysteine (Cys) sulfhydryl groups of proteins or peptides, such as glutathione (GSH) [16]. Conjugation with the tripeptide GSH (γGlu-Cys-Gly) is an important detoxification reaction in phase II drug metabolism [17]. The reaction increases polarity and thus facilitates excretion of xenobiotics [17]. The involved glutathione-*S*-transferases (GSTs), together with uridine diphosphate glucuronosyl transferases (UGTs), sulfotransferases (SULTs) are the most involved phase II enzymes in the metabolism of clinically used drugs besides *N*-acetyl transferases (NATs) and thiopurine-*S*-methyltransferase (TPMT) [17]. Apart from being catalyzed by GST, GSH conjugation may also occur spontaneously if the reactivity of the xenobiotic is high enough, which has been shown for both helenalin and 11α,13-dihydrohelenalin. For both STLs, a strong spontaneous reaction with GSH was reported. The resulting GSH mono-conjugates of helenalin and 11α,13-dihydrohelenalin and the GSH di-conjugate of helenalin were found to inhibit GST from horse liver, while free helenalin and 11α,13-dihydrohelenalin showed no inhibitory activity [18]. In another report, helenalin was described to inhibit hepatic and renal GST from rats, not considering the possibility that this inhibition could be caused by the GSH conjugates formed under the assay conditions [19]. In the same study, inhibitory effects on renal GSH reductase and hepatic superoxide dismutase activity were observed, while the activity of rat renal and hepatic GSH peroxidase was reported to increase. In addition, effects of Arnica STL on phase I enzymes were described. Helenalin inhibited rat and mouse cytochrome P450 monooxygenases (CYP 450) activity [19,20]. These findings are in agreement with the observation that several STLs affect enzymes involved in cell homeostasis, particularly in relation to redox status and detoxification of harmful xenobiotics [21].

Besides these effects on metabolic enzymes and the formation of GSH conjugates, the metabolism of Arnica STLs has, to the best of our knowledge, not been investigated systematically and in detail up to now. It was shown that Arnica STLs penetrate through human epidermal membranes after dermal application [22]. After absorption into the blood, xenobiotics are distributed to organs, e.g., the liver, and both metabolized and excreted [23]. Before the treatment of open lesions of human CL patients with Arnica tincture, the possible metabolism of the active constituents should be investigated in a detailed manner. Therefore, in vitro metabolism experiments with liver microsomes of rat, pig and human origin were performed. Liver microsomes are fragments of the endoplasmic reticulum of hepatic cells, which are prepared by homogenization of liver followed by centrifugation steps [24]. Since liver microsomes contain many drug-metabolizing enzymes, such as CYP 450, flavin monooxygenases, epoxide hydrolase and UGT, they are widely used as in vitro metabolism model [24].

The aim of this study was the detailed identification of metabolites formed from two representatives of the main STLs from Arnica tincture.

## 2. Results and Discussion

Arnica tincture contains a complex mixture of some 20 different ester derivatives of helenalin (H) and 11α,13-dihydrohelenalin (DH). Therefore, we chose two representative esters for the present investigation. Helenalin acetate (Hac) and 11α,13-dihydrohelenalin acetate (DHac) (cf. Figure 1) were selected covering the two different types of main constituents within the STL-mixture. With these two Arnica STLs, different phase I and phase II metabolism experiments were performed. Furthermore, a combined phase I and phase II experiment with physiological GSH concentration was carried out. This combined experiment best describes the physiological competition of phase I and phase II reactions. Therefore, the main focus of this article is on the results of this latter experiment. After different incubation times with a nicotinamide adenine dinucleotide phosphate (NADPH) regenerating system, GSH and GST, 39 metabolites were detected. Sample analysis was performed with an ultra-high-performance liquid chromatography coupled to a quadrupole time-of-flight high-resolution mass spectrometer with electrospray ionization in positive mode (UHPLC-(+ESI)-QqTOF MS). Identification of the metabolites is based on accurate mass measurements and fragmentation experiments. The main metabolic pathway of the Arnica STLs was found to be GSH conjugation, followed by subsequent reactions. Moreover, we also report on the formation of various phase I metabolites of the STLs.

### 2.1. Phase I Metabolism

For the formation of phase I metabolites, an NADPH regenerating system was added to the liver microsomes as described by Sohl et al. [25]. Activity of enzymes in the microsomes mix and functionality of the NADPH-regenerating system was ensured with the positive control harmane. In each experimental setup, metabolic reactions were terminated by adding of acetonitrile after different incubation times (30 min, 60 min, 120 min and 240 min) to investigate the progress of metabolism over time and ensure detection of intermediate as well as final products. Then, samples were analyzed with a UHPLC-(+ESI)-QqTOF MS system. A matrix control sample containing dimethyl sulfoxide (DMSO) instead of the analytes was analyzed and used for background subtraction. Chromatograms processed in this way are marked as difference chromatograms. Appendix A show the difference chromatograms of DHac and Hac after incubation with rat liver microsomes (RLM), pig liver microsomes (PLM) and human liver microsomes (HLM) for 30 min, 60 min, 120 min and 240 min, respectively. The main metabolites of DHac and Hac were typical phase I metabolites, such as water conjugates and hydroxides (cf. combined phase I and II experiments, Section 2.3). Furthermore, small amounts of GSH and Cys present in the microsomes mix led to a small proportion of GSH- and Cys conjugates (DHac-GSH, Hac-GSH and Hac-Cys) after 30 min of incubation.

With longer incubation times, degradation of GSH conjugates led to Hac-CysGly (with RLM) and further to Hac-Cys (with RLM). Another difference, depending on the species, was observed concerning the metabolism of the exocyclic methylene group of Hac. After incubation with PLM a large proportion of this double bond was reduced and thus Hac was converted to DHac. This reduction was strong with PLM, weak with RLM and, interestingly, not observed with HLM. On the other hand, very small amounts of Hac were formed from DHac by incubation with PLM. Surprisingly, no signs of ester hydrolysis could be detected in any of the experiments, despite being thoroughly sought for.

### 2.2. Phase II Metabolism (Glucuronidation and Sulfation)

Based on the formed phase I metabolites described in Section 2.1, possible phase II reactions of the major phase II enzymes were considered. The two predominant phase II reactions of clinically used drugs, glucuronidation and sulfation, might take place at free hydroxy groups of the hydroxides and water conjugates. The in-depth study of GSH conjugation is reported in Section 2.3. Independently, glucuronidation and sulfation were investigated by incubation of Hac and DHac with PLM and uridine 5′-diphosphoglucuronic acid (UDPGA) or pig liver cytosol (PLC), dithiothreitol (DTT) and 3′-phosphoadenosine-5′-phosphosulfate (PAPS), respectively. These experiments were performed with native Hac and DHac and furthermore with the corresponding incubation mixtures after phase I metabolism (cf. Section 2.1). Although the latter contained metabolites with free hydroxy groups (hydroxides and water conjugates), no glucuronides or sulfates were formed (data not shown). The positive control 7-hydroxy coumarin was metabolized to its glucuronide and sulfate as expected, confirming active UGT and SULT in the reaction mixtures.

### 2.3. Combined Phase I and Phase II Metabolism

As mentioned above, the phase I experiments showed formation of GSH conjugates with small amounts of residual GSH in the microsomes mix. However, most of the intracellular GSH is removed by centrifugation during the preparation of the microsomes so that their GSH concentration is very low. To mimic physiological conditions, GSH as well as GST were added in appropriate near-physiological concentrations [26,27]. The occurrence of other major phase II reactions was already excluded (cf. Section 2.2) so that no further enzymes or cofactors were added. In this experiment (which mimics, but does not necessarily fully represent, physiological conditions), phase I and phase II reactions competed for the reactive sites of the STLs. Figure 2 shows the difference chromatograms of DHac after incubation with the NADPH regenerating system, GSH, GST and RLM, PLM or HLM for 30 min, 60 min, 120 min, 240 min and 24 h, respectively. The reactant (DHac) elutes at a retention time of 6.9 min. An overview of all observed DHac metabolites detected in the different experimental setups is given in Table 1. In total, 15 DHac metabolites were identified, including 2 GSH conjugates (**1**, **2**) 2 hydroxylated GSH conjugates (**3**, **4**), 2 water conjugates (**5**, **6**), 1 methylated water conjugate (**7**), 4 mono-hydroxides (**8**, **9**, **10**, **11**), 1 di-hydroxide (**12**), 2 hydroxylated water conjugates (**13**, **14**) and 1 dehydrogenated metabolite (**15** = Hac).

Regardless of the species, the main metabolites of DHac after 30 min of incubation were GSH conjugates (**1**, **2**). The formation of two metabolites can be explained with the formation of DHac-2α-*S*-GSH, as well as that of DHac-2β-*S*-GSH. The different extents of formation of both stereoisomers might be explained by steric interactions, which would be in accordance with previous observations [16,28]. With increasing incubation times, the GSH conjugates diminished, and typical phase I metabolites predominated. However, some species dependent differences were observed. In case of PLM and HLM, first phase I metabolites were already formed after 30 min of incubation, whereas in case of RLM this development was only observed after 24 h of incubation. Among them, water conjugates (**5**, **6**), hydroxides (**8**, **10**, **12**) and combinations of both (**13**, **14**) were formed. Reversibility of STL-GSH conjugation as observed in this experiment is in accordance with the results in the literature [16]. Hydroxylation is likely catalyzed by CYP 450 which are involved in phase I metabolism of approximately 75% of pharmaceuticals [29]. Carbons in α-position to functional groups are positions that are likely to be hydroxylated. Furthermore, a preference of CYP 450 for tertiary rather than secondary and primary carbons was reported [30]. The most probable position for water addition is the Michael acceptor (C-2 conjugation), as in many biological reactions [31]. Another possibility is hydration at the lactone moiety [32]. Structure elucidation of the metabolites was based on MS/MS fragmentation experiments as well as consideration of (bio)chemically feasible reactions. Exemplarily, MS/MS spectra of DHac and its most abundant metabolites are shown in Figure 3. A detailed description for all metabolites can be found in Appendix A.

The first step in ESI-MS fragmentation of DHac is ester cleavage followed by dehydration at the resulting hydroxy group (corresponding to a loss of acetic acid, C_2_H_4_O_2_, 60.0265 Da) leading to the fragment *m*/*z* 247.1358 ([M+H-C_2_H_4_O_2_]^+^). Additional dehydration at the lactone moiety results in fragment *m*/*z* 229.1239 ([M+H-C_2_H_4_O_2_-H_2_O]^+^). Further fragmentation at the lactone moiety leads to the fragments *m*/*z* 201.1289 ([M+H-C_2_H_4_O_2_-H_2_O-CO]^+^) and *m*/*z* 173.1058 ([M+H-C_2_H_4_O_2_-H_2_O-CO-C_2_H_4_]^+^). The latter results in *m*/*z* 145.0997 by CO (27.9949 Da) cleavage (cf. Figure 3).

For the DHac-GSH conjugate (**1**; [M+H]^+^ at *m*/*z* 614. 2493), the typical DHac fragment *m*/*z* 247.1394 ([M+H-GSH-C_2_H_4_O_2_]^+^) is observed. The fragment *m*/*z* 539.2262 might result from neutral loss of glycine (C_2_H_5_NO_2_, 75.0325 Da) from the DHac-GSH conjugate. Other fragments of this metabolite result from the GSH moiety: *m*/*z* 308.0885 represents protonated free GSH (C_10_H_18_N_3_O_6_S^+^, 308.0911 Da), whereas *m*/*z* 179.0488 is formed by neutral loss of glutamic acid (C_5_H_9_NO_4_, 129.0426 Da) from GSH.

The hydroxylated metabolite (**10**; [M+H]^+^ at *m*/*z* 323.1492) is cleaved like DHac (neutral loss of acetic acid) resulting in *m*/*z* 263.1226 ([M+H-C_2_H_4_O_2_]^+^), which, after neutral loss of CO gave the fragment *m*/*z* 235.1370. After dehydration of both aforementioned fragments, *m*/*z* 245.1218 ([M+H-C_2_H_4_O_2_-H_2_O]^+^) and *m*/*z* 217.1245 ([M+H-C_2_H_4_O_2_-H_2_O-CO]^+^) result, respectively. Cleavage at the lactone moiety then leads to *m*/*z* 199.1122 ([M+H-C_2_H_4_O_2_-2H_2_O-CO]^+^) and *m*/*z* 175.1071 ([M+H-C_2_H_4_O_2_-H_2_O-CO-C_2_H_2_O]^+^), respectively. The latter is only formed if the hydroxy group is cleaved with C-11 and C-13 as hydroxyacetylene (C_2_H_2_O, 42.0106 Da). In comparison of both carbons (C-11 and C-13), the tertiary carbon C-11 is much more likely hydroxylated than the primary carbon C-13 [26].

Fragments of the combined hydroxide and water conjugate (**14**; [M+H]^+^ at *m*/*z* 341.1614), are *m*/*z* 281.1337 ([M+H-C_2_H_4_O_2_]^+^) resulting from loss of acetic acid, *m*/*z* 263.1263 ([M+H-C_2_H_4_O_2_-H_2_O]^+^) after further dehydration, *m*/*z* 245.1130 ([M+H-C_2_H_4_O_2_-2H_2_O]^+^) ensuing from a second dehydration, and the corresponding fragments after neutral loss of CO: *m*/*z* 253.1397, *m*/*z* 235.1310 and *m*/*z* 217.1202, respectively. Analogous to (**10**), the fragment *m*/*z* 175.1109 results from cleavage of the lactone moiety and shows that hydroxylation occurred at C-11.

To evaluate metabolic stability, the peak area of the unaltered analyte (negative control) was compared with the peak area after incubation. In case of DHac, a substantial decrease in the reactant was observed with remaining portions of unaltered analyte of 62.3% (RLM), 66.2% (PLM) and 50.4% (HLM) after 30 min and 53.5% (RLM), 35.2% (PLM) and 9.5% (HLM) after 24 h of incubation. In case of extensive metabolism, identification of the subsequent metabolites is important for risk evaluations.

Analysis of DHac incubated with GSH and the NADPH regenerating system but without GST and microsomes allows conclusions about the involvement of enzymes in the observed reactions. Metabolites detected in this approach (cf. Appendix A) were formed by spontaneous (uncatalyzed) chemical rather than enzymatic reactions. For DHac, one of the water conjugates (**5**) was detected and thus obviously formed spontaneously. Furthermore, the main metabolites, GSH conjugates (**1**) and (**2**), were formed in even higher amounts in the absence of GST and microsomes compared with the incubation with enzymes (cf. Appendix A). This result appears unexpected, since GST normally catalyzes the conjugation of xenobiotics with GSH. However, the reactivity of the cyclopentenone moiety of H and DH has previously been shown to be rather high and an involvement of GST in the formation of their GSH conjugates was questioned [18]. A possible explanation for the lower amount of GSH conjugates (**1**) and (**2**) could be metabolism/degradation of GSH resulting in lower levels of GSH available for STL conjugation.

Analogous metabolism experiments were carried out with Hac, which, in addition to the cyclopentenone unit analogous to DHac, contains another Michael acceptor, the exocyclic methylene group as part of an α-methylene-γ-lactone moiety. Figure 4 shows the difference chromatograms of Hac incubated with the combined phase I and II approach and RLM, PLM or HLM, respectively. These experiments yielded 24 metabolites in total, including GSH mono- and di-conjugates (**16**, **17**, **19**, **20**, **21**, **22**), degradation products of the former (**18**, **26**, **27**, **32**), water conjugates (**33, 34**), a methylated water conjugate (**35**) and combinations thereof (**23**, **24**, **25**, **28**, **29**, **30**, **31**). Furthermore, hydroxides (**36**, **37**), 1 water conjugated hydroxide (**38**) and 1 hydrogenated metabolite (**39** = DHac) were formed.

After 30 min of incubation, GSH mono- and di-conjugates of Hac predominated (**16**–**17**, **19**–**22**). Moreover, a GSH-water conjugate (**23**), methylated GSH-water conjugates (**24**, **25**) and a Hac-GSH-CysGly conjugate (**18**) were detected. The latter is likely to be formed by cleavage of glutamic acid from a GSH di-conjugate. For the GSH mono-conjugates, the corresponding degradation product Hac-CysGly (**26**, **27**) is also found after longer incubation times. The Hac-CysGly conjugates were the most abundant metabolites after 24 h for all investigated species. Furthermore, water conjugates of these metabolites were identified (**28**, **29**, **30**, **31**).

Similar to DHac, the amount of water conjugates (**33**, **34**) increased with longer incubation times. In the pure phase I experiments with PLM and RLM without GST and GSH supplement, hydrogenation of the exocyclic methylene group of Hac led to the formation of DHac (cf. Section 2.1). A comparable hydrogenation in the metabolism of another STL (alantolactone) leading to 11,13-dihydroalantolactone was reported by Yao et al., who observed this transformation when investigating bacterial metabolites of the STL [33].

This reduction reaction was dominant in the absence of physiological GSH concentrations but could be neglected in the combined phase I and phase II experiment because GSH conjugation strongly predominated.

Reversibility of GSH conjugation is corroborated by this experiment but is weaker at the exocyclic methylene group than at the cyclopentenone structure. This can be concluded, as GSH mono-conjugates of Hac are still detectable after 24 h of incubation (cf. Figure 4) whereas no GSH conjugate of DHac was detected after 24 h (cf. Figure 2). This finding is in agreement with a previous report in which this difference in reversibility was concluded based on results on the influence of GSH concentrations on the cytotoxicity of helenalin and related STLs [34]. Degradation of GSH conjugates to CysGly conjugates as seen in this experiment is part of the *N*-acetyl cysteine (NAC) pathway [35]. Further steps in this pathway are loss of Gly leading to Cys conjugates and *N*-acetylation forming mercapturic acids. The former was found for Hac in the phase I experiment without GSH and GST supplement (cf. Appendix A, PLM, 60–240 min). As GSH conjugates are more stable at the exocyclic methylene group, it can be concluded that the degradation products resulting from the loss of one or two amino acids are derived from the conjugates at this position. Besides the degradation of GSH conjugates to Cys conjugates within the NAC pathway, the formation of Cys conjugates is also known to occur directly with free Cys. In this process, Cys is much more likely to react with the exocyclic methylene group than with the cyclopentenone moiety of helenalin [28]. This conjugation was also observed in the phase I experiment without GSH and GST supplement (cf. Appendix A, HLM, 30 min). Since only one Cys conjugate (**32**) was found, it can be assumed that Cys was added at the exocyclic methylene group. In agreement with the previous literature [28], this would also explain why no Cys conjugate was formed with DHac, which has no exocyclic methylene group. An overview of all detected metabolites of Hac is given in Table 2.

ESI-MS fragmentation of Hac ([M+H]^+^ at *m*/*z* 305.1582) is identical to the fragmentation of DHac (cf. Figure 5). In the first step, loss of acetic acid thus leads to *m*/*z* 245.1207 ([M+H-C_2_H_4_O_2_]^+^). Further dehydration results in *m*/*z* 227.1096 ([M+H-C_2_H_4_O_2_-H_2_O]^+^). Fragments at *m*/*z* 199.1128 ([M+H-C_2_H_4_O_2_-H_2_O-CO]^+^) and at *m*/*z* 173.0958 ([M+H-C_2_H_4_O_2_-H_2_O-CO-C_2_H_4_]^+^) derive from cleavage at and of the lactone moiety. Fragmentation of the GSH di-conjugate (**16**; [M+H]^+^ at *m*/*z* 919.3190) leads to *m*/*z* 790.2652 ([M+H-C_5_H_9_NO_4_]^+^) by neutral loss of glutamic acid (C_5_H_9_NO_4_, 129.0426 Da).

Similarly, fragmentation of the GSH mono-conjugate (**21**; [M+H]^+^ at *m*/*z* 612.2322) leads to *m*/*z* 483.1897 by neutral loss of glutamic acid and fragment *m*/*z* 537.2004 by neutral loss of glycine (C_2_H_5_NO_2_, 75.0325 Da), respectively.

Hac-CysGly (**26**; [M+H]^+^ at *m*/*z* 483.1838) shares the core fragment *m*/*z* 245.1205 with the parent compound Hac. Fragment *m*/*z* 423.1582 ([M+H-C_2_H_4_O_2_]^+^) is formed by neutral loss of acetic acid. Additional loss of ammonia (NH_3_, 17.0265 Da) leads to *m*/*z* 406.1329 ([M+H-C_2_H_4_O_2_-NH_3_]^+^).

Investigation of the metabolic stability of Hac after 30 min revealed a substantial decrease in the reactant with remaining portions of 30.3% (RLM), 29.3% (PLM) and 12.8% (HLM), respectively. This degradation tendency continued and after 24 h of incubation only 8.6% (RLM), 12.2% (PLM) and 5.3% (HLM) of the reactant remained. These results highlight the importance of metabolic investigations for safety assessments.

Analogous to the observation with DHac, GSH mono- and di-conjugates of Hac were formed in high concentrations in the absence of enzymes (cf. Appendix A).

An overview of the postulated chemical structures of all detected metabolites of DHac and Hac is shown in Figure 6. Identification of the metabolites is based on accurate mass, shifts in retention time (i.e., polarity) and plausibility of observed MS/MS fragmentation as well as feasibility in mammalian metabolism. A detailed description for every metabolite can be found in Appendix A.

The metabolism of Hac and DHac shows similarities with that of other STLs, e.g., alantolactone, isoalantolactone, costunolide and dehydrocostus lactone [36,37]. In a review on the pharmacokinetics of different STLs, phase I and phase II metabolism of STLs were described to be extensive, which agrees with our results [38]. For example, the phase I metabolic reactions hydroxylation, hydration and desaturation, as well as the phase II metabolic reactions GSH conjugation, Cys conjugation and methylation, were observed in our experiments and also in a pharmacokinetic study of costunolide and dehydrocostus lactone by Peng et al. [36].

For alantolactone and isoalantolactone, the predominant metabolic pathway was described to be GSH conjugation even in the absence of enzymes, just as in our observations [37]. Related to this, the exocyclic methylene group of the α,β-unsaturated lactone moiety of alantolactone and isoalantolactone was described to be their major metabolic target [37]. This Michael acceptor system was also highlighted as a major metabolism region in the above-mentioned review on the pharmacokinetics of different STLs [38]. Our results are consistent with this conclusion, because Michael acceptor systems are also the major metabolic targets in case of Hac and DHac.

## 3. Materials and Methods

### 3.1. Chemicals and Materials

All chemicals were purchased from Merck KGaA (Darmstadt, Germany), VWR International GmbH (Langenfeld, Germany), Carl Roth GmbH and Co. KG (Karlsruhe, Germany) or Thermo Fisher Scientific (Schwerte, Germany). Water was purified by an ELGA PURELAB system (Veolia Water Technologies, Celle, Germany). Mixed gender HLM of 150 donors were purchased from Corning (Wiesbaden, Germany). RLM, PLM and PLC were prepared according to established protocols [39]. Determination of protein content was carried out as described by Bradford [40]. Liver cytosol and liver microsomes were stored at −80 °C prior to usage. Hac and DHac were isolated from (385 g) dried and powdered *Arnica montana* flowerheads by Soxhlet extraction with dichloromethane. Afterwards, flash chromatography on silica gel and ambient pressure column chromatography was performed with hexane and ethylacetate. From various subfractions, Hac (89 mg) and DHac (7 mg) were obtained by preparative HPLC (Jasco, Groß-Umstadt, Germany) with preparative reverse phase column Reprosil 100 C18 (5 μm, 250 mm, 20 mm, Macherey-Nagel, Düren, Germany). The identity and purity (>95%) of both STLs was assessed by ^1^H NMR spectroscopy.

### 3.2. Incubation with Liver Microsomes or Liver Cytosol

Phase I metabolism, glucuronidation and sulfation experiments were accomplished based on established protocols [25,41,42]. Briefly, the test compounds were incubated in duplicate with different incubation mixtures, as described in the following sections. For each experiment, different control samples were prepared, including positive control, matrix control, stability control and negative control. The conversion of the positive controls harmane (phase I) or 7-hydroxycumarin (phase II) to their known metabolites confirmed enzymatic activity in the incubation mixtures. The matrix control with DMSO instead of a test compound was analyzed to subtract matrix signals. A stability control of the test compound was incubated without enzymes to differentiate between enzymatic and spontaneous reactions. As negative controls, test compounds were incubated in water without supplements.

#### 3.2.1. Phase I Metabolism

The test compounds were transferred to reaction vessels and evaporated to dryness. Then, 100 µL of the incubation mixture were added to the analytes (0.1 mM) and samples were incubated at 37 °C on a laboratory shaker (150 rpm) in the dark. Incubation mixtures contained RLM, PLM or HLM (5 mg protein/mL) and an NADPH-regenerating system (10 mM glucose-6-phosphate, 0.5 mg/mL NADP^+^, 2 U/mL glucose-6-phosphate dehydrogenase) in potassium phosphate buffer (0.1 mM, pH 7.4). In the combined phase I and II metabolism experiments, the phase I incubation mixture was supplemented with GSH (0.5 mM) and GST (0.125 mg/mL; ≥25 units/mg protein). After different incubation times (30 min, 60 min, 90 min, 120 min, 240 min or 24 h), ice-cold acetonitrile (200 µL) was added to precipitate proteins and stop the reactions. After vortex mixing and centrifugation for 5 min at 14,000× *g* and 4 °C, 150 µL of the supernatant were diluted with 850 µL of purified water. The samples were analyzed with an UHPLC-ESI-qQTOF MS system in positive ionization mode.

#### 3.2.2. Phase II Metabolism

Phase II metabolism experiments were performed similar to phase I metabolism experiments but with different incubation mixtures. For glucuronidation experiments, test compounds (0.1 mM) were incubated with UDPGA (5 mM), PLM (1 mg protein/mL) and magnesium chloride (5 mM) in tris buffer (0.05 M, pH 7.4). For sulfation approaches, incubation was carried out with DTT (1.0 mM), PAPS (0.08 mg/mL) and PLC (10 mg protein/mL). Further sample preparation was identical to the phase I metabolism experiments (cf. Section 3.2.1) but UHPLC-ESI-qQTOF MS analysis was carried out in negative ionization mode.

Additional phase II glucuronidation and sulfation experiments were performed with supernatants of the phase I experiment. Therefore, 50 µL of phase I supernatants (of Hac and DHac samples) were transferred to reaction vessels and evaporated to dryness. Samples were incubated as described previously, but in a final volume of 50 µL. After 240 min, ice-cold acetonitrile (200 µL) was added, samples were vortex mixed and centrifuged, as described before. Prior to UHPLC-QqTOF MS analysis, 250 µL of the supernatant were evaporated to dryness and dissolved in 100 µL acetonitrile (5%) to concentrate samples.

### 3.3. UHPLC-QqTOF MS Analysis

UHPLC-QqTOF MS analysis of samples was performed with an Ultimate 3000 UHPLC system (Dionex, Sunnyvale, CA, USA) equipped with an Ultimate 3000 RS Diode Array Detector (Dionex, Sunnyvale, CA, USA) coupled to a micrOTOF-Q II quadrupole time-of-flight mass spectrometer (Bruker Daltonics, Billerica, MA, USA) with electrospray ionization in positive or negative ionization mode. Chromatographic separation was carried out with an Acclaim RSLC 120 C18 analytical column (particle size 2.2 μm, diameter 2.1 mm, length 0.1 m; Dionex, Sunnyvale, CA, USA) at 40 °C. A binary gradient of water (A) and acetonitrile (B) with 0.1% formic acid each (*v*/*v*), a flow rate of 0.4 mL/min and an injection volume of 10 µL was used. The portion of mobile phase B started with 5% B, increased to 100% B within 9.5 min, was held at 100% B until 15.0 min, returned to 5% B until 15.1 min, and stayed at 5% B until 20.0 min. For all metabolites, product ion scans with the metabolite as precursor were performed with this method. For MS/MS analysis of minor metabolites, an increased injection volume of 100 µL and a longer separation method was used with the following gradient: 0.0 min, 5% B; 36.0 min, 100% B; 47.0 min, 100% B; 48.0 min 5% B; 55.0 min, 5% B.

For MS and MS/MS analysis, nitrogen was used as nebulizer gas (3.5 bar) and as dry gas (9 L/min, 200 °C). Capillary voltage was set at +4500 V in positive ionization mode, and −3500 V in negative ionization mode. For full transition experiments, a mass range of *m*/*z* 50–1500 was chosen. For MS/MS analysis, a collision energy of 20 eV and nitrogen as collision gas was used. Mass calibration was achieved with a solution of sodium formate (10 mM) in isopropanol/water/formic acid/sodium hydroxide (50/50/0.2/1, *v*/*v*).

For data acquisition and operation of the system, the software packages Chromeleon (Dionex, Sunnyvale, CA, USA) and OTOF control (Bruker Daltonics, Billerica, MA, USA) were used. Data processing was performed with the software packages Data Analysis (Bruker Daltonics, Billerica, MA, USA) and Metabolite Detect (Bruker Daltonics, Billerica, MA, USA).

## 4. Conclusions

In this article, we report, for the first time, in detail, on the metabolism of helenalin acetate and 11α,13-dihydrohelenalin acetate as typical examples of the STLs present in *Arnica* flowers and tincture. In total, 24 metabolites of helenalin acetate and 15 metabolites of 11α,13-dihydrohelenalin acetate were identified based on MS analysis after incubation with rat, pig or human liver microsomes. For both STLs, a high extent of metabolism was observed. We demonstrate that the major metabolic route is glutathione conjugation followed mainly by degradation reactions, such as the loss of glutamic acid. Typical phase I reactions play a minor role in the metabolism if GSH is present in physiological concentrations. Due to the reactivity of the STLs, GSH conjugates are also formed spontaneously, i.e., without the contribution of GST. This confirms that STLs should be able to react also with SH groups of proteins, as described before, which would be necessary to explain their pharmacological effects. With regard to toxicological assessment, glutathione conjugates as the main metabolites are of no toxicological concern [43]. It is assumed that the degradation products formed by loss of amino acids are harmless as well. These findings support the traditional use of Arnica tincture and new applications such as treatment of CL lesions. Detailed studies on the dermal absorption of STLs from Arnica tincture are in progress.

## Figures and Tables

**Figure 1 metabolites-12-00088-f001:**
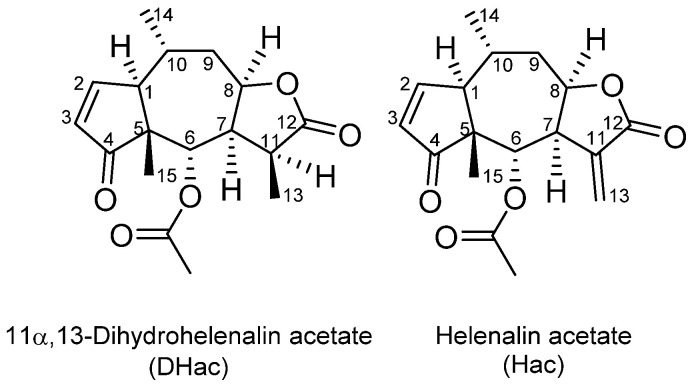
Chemical structures of the STLs under study.

**Figure 2 metabolites-12-00088-f002:**
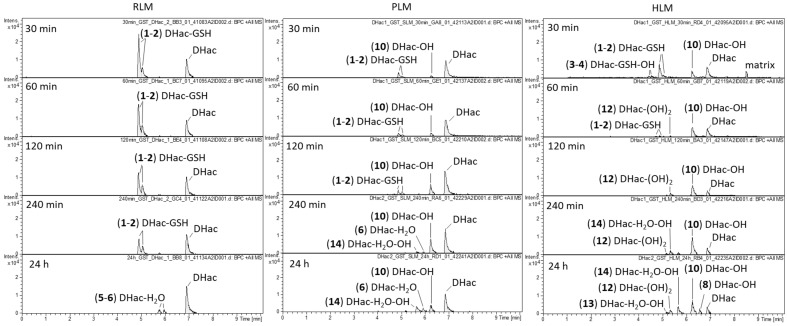
Difference chromatograms of DHac after incubation with RLM, PLM or HLM, NADPH regenerating system, GST and GSH for 30 min, 60 min, 120 min, 240 min and 24 h, respectively.

**Figure 3 metabolites-12-00088-f003:**
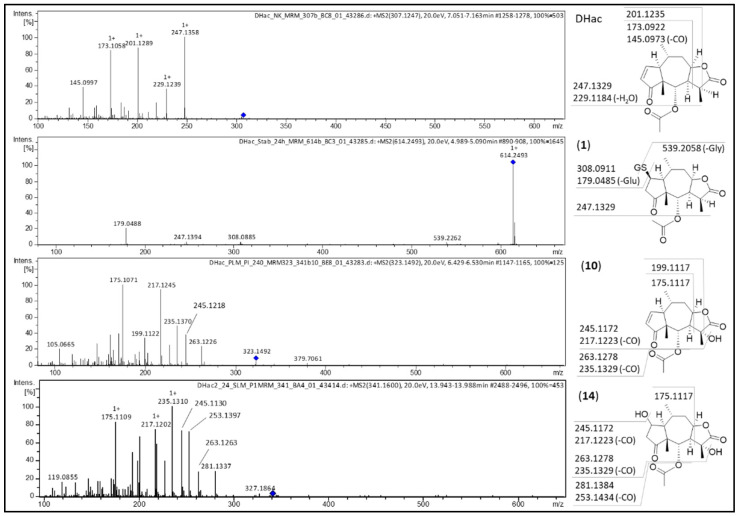
MS/MS spectra of DHac and its most abundant metabolites DHac-GSH (**1**), DHac-OH (**10**) and DHac-H_2_O-OH (**14**) with postulated fragmentation sites. At the structural diagrams the calculated *m*/*z* values are given.

**Figure 4 metabolites-12-00088-f004:**
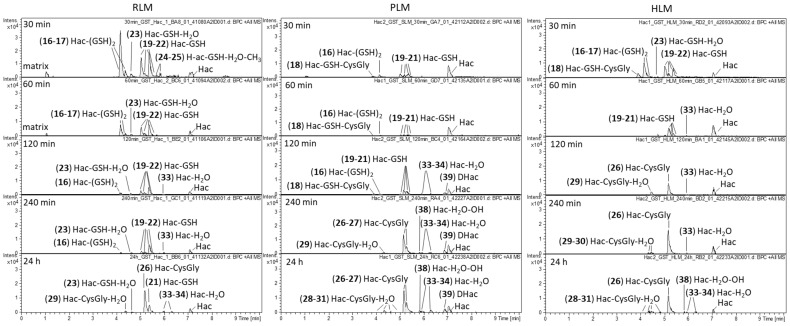
Difference chromatograms of Hac after incubation with RLM, PLM or HLM, NADPH regenerating system, GST and GSH for 30 min, 60 min, 120 min, 240 min and 24 h, respectively.

**Figure 5 metabolites-12-00088-f005:**
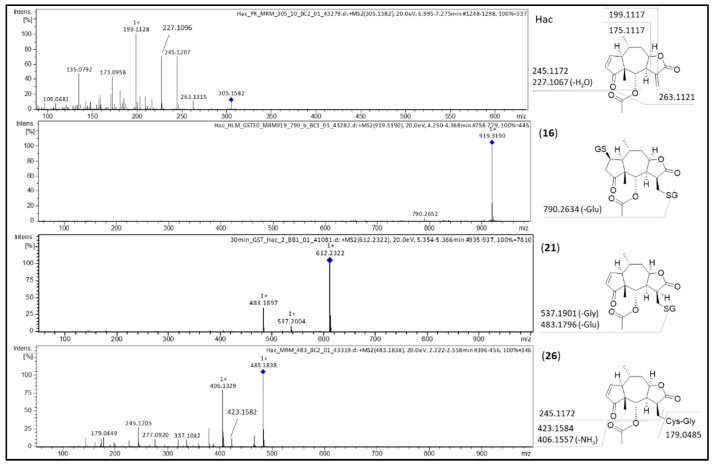
MS/MS spectra of Hac and its most abundant metabolites Hac-(GSH)_2_ (**16**), Hac-GSH (**21**) and Hac-CysGly (**26**) with postulated fragmentation sites. At the structural diagrams the calculated *m*/*z* values are given.

**Figure 6 metabolites-12-00088-f006:**
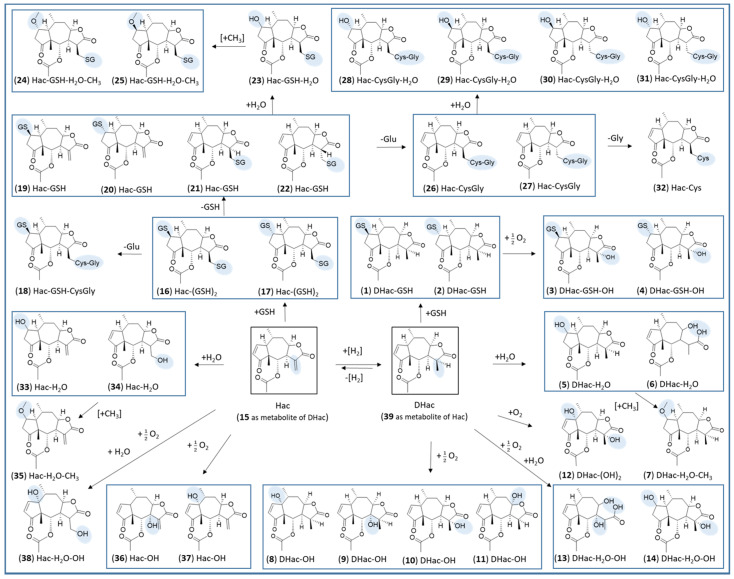
Postulated chemical structures of detected phase I and phase II metabolites of DHac and Hac. Boxes around groups of metabolites indicate the formation by the same biotransformation reaction. Please note that the assignment of relative configuration for the conjugate metabolites based on MS is tentative but based on chemical evidence (cf. Appendix A).

**Table 1 metabolites-12-00088-t001:** Suggested biotransformation of DHac based on phase I and phase II metabolism experiments. Metabolites detected by UHPLC-(+ESI)-QqTOF MS are grouped by similar biotransformation reactions and characterized by their retention time (Rt), elemental formula, measured protonated molecule [M+H]^+^ and mass deviations (Δm—measured/calculated mass).

No.	Metabolite	Biotransformation	Rt (min)	Formula	[M+H]^+^ (*m*/*z*)	Δm (mDa) ^a^	Δm (ppm) ^a^
**1**	DHac-GSH	GSH conjugation	4.9	C_27_H_39_N_3_O_11_S	614.2379	0.0	0.0
**2**	DHac-GSH	GSH conjugation	5.1	C_27_H_39_N_3_O_11_S	614.2372	−0.7	−1.1
**3**	DHac-GSH-OH	GSH conjugation, hydroxylation	4.4	C_27_H_39_N_3_O_12_S	630.2346	1.9	3.0
**4**	DHac-GSH-OH	GSH conjugation, hydroxylation	4.5	C_27_H_39_N_3_O_12_S	630.2333	0.6	1.0
**5**	DHac-H_2_O	water addition	5.8	C_17_H_24_O_6_	325.1648	0.2	0.6
**6**	DHac-H_2_O	water addition	6.0	C_17_H_24_O_6_	325.1643	−0.3	−0.9
**7**	DHac-H_2_O-CH_3_	water addition, *O*-methylation	7.7	C_18_H_26_O_6_	339.1822	2.0	5.9
**8**	DHac-OH	hydroxylation	5.1	C_17_H_22_O_6_	323.1482	−0.8	−2.5
**9**	DHac-OH	hydroxylation	5.8	C_17_H_22_O_6_	323.1498	0.8	2.5
**10**	DHac-OH	hydroxylation	6.3	C_17_H_22_O_6_	*323.1500*	1.0	3.1
**11**	DHac-OH	hydroxylation	6.6	C_17_H_22_O_6_	323.1484	−0.6	−1.9
**12**	DHac-(OH)_2_	hydroxylation	5.3	C_17_H_22_O_7_	339.1452	1.4	4.1
**13**	DHac-H_2_O-OH	water addition, hydroxylation	5.2	C_17_H_24_O_7_	341.1583	−1.2	−3.5
**14**	DHac-H_2_O-OH	water addition, hydroxylation	5.7	C_17_H_24_O_7_	341.1614	1.9	5.6
**15**	Hac	dehydrogenation	7.1	C_17_H_20_O_5_	305.1373	−1.1	−3.6

^a^ Mass accuracy of the used QqTOF MS system (Bruker micrOTOF-Q II) is specified as ±2 mDa (*m*/*z* < 400) and ±5 ppm (*m*/*z* > 400) for signals with intensity > 10,000 counts). The calculated mass deviations are within these specifications.

**Table 2 metabolites-12-00088-t002:** Suggested biotransformation of Hac based on phase I and phase II metabolism experiments. Metabolites detected by UHPLC-(+ESI)-QqTOF MS are grouped by similar biotransformation reactions and characterized by their retention time (Rt), elemental formula, measured protonated molecule [M+H]^+^ and mass deviations (Δm—measured/calculated mass).

No.	Metabolite	Biotransformation	Rt (min)	Formula	[M+H]^+^ (*m*/*z*)	Δm (mDa) ^a^	Δm (ppm) ^a^
**16**	*Hac* *-(GSH)_2_*	GSH conjugation	4.2	C_37_H_54_N_6_O_17_S_2_	919.3017	4.3	4.7
**17**	Hac-(GSH)_2_	GSH conjugation	4.4	C_37_H_54_N_6_O_17_S_2_	919.3043	1.7	1.8
**18**	Hac-GSH-CysGly	GSH conjugation; cleavage of Glu	3.9	C_32_H_47_N_5_O_14_S_2_	790.2672	3.8	4.8
**19**	Hac-GSH	GSH conjugation	5.0	C_27_H_37_N_3_O_11_S	612.2245	2.3	3.8
**20**	Hac-GSH	GSH conjugation	5.2	C_27_H_37_N_3_O_11_S	612.2204	−1.8	−2.9
**21**	Hac-GSH	GSH conjugation	5.4	C_27_H_37_N_3_O_11_S	612.2231	0.9	1.5
**22**	Hac-GSH	GSH conjugation	5.6	C_27_H_37_N_3_O_11_S	612.2217	0.5	0.8
**23**	Hac-GSH-H_2_O	GSH conjugation; water addition	4.6	C_27_H_39_N_3_O_12_S	630.2311	−1.6	−2.5
**24**	Hac-GSH-H_2_O-CH_3_	GSH conjugation; water addition; *O*-methylation	5.6	C_28_H_41_N_3_O_12_S	644.2513	2.9	4.5
**25**	Hac-GSH-H_2_O-CH_3_	GSH conjugation; water addition; *O*-methylation	5.7	C_28_H_41_N_3_O_12_S	644.2460	−2.4	−3.7
**26**	Hac-CysGly	GSH conjugation, cleavage of Glu	5.1	C_22_H_30_N_2_O_8_S	483.1814	1.8	3.7
**27**	Hac-CysGly	GSH conjugation, cleavage of Glu	5.3	C_22_H_30_N_2_O_8_S	483.1814	1.8	3.7
**28**	Hac-CysGly-H_2_O	GSH conjugation, cleavage of Glu	4.0	C_22_H_32_N_2_O_9_S	501.1885	−1.6	−3.2
**29**	Hac-CysGly-H_2_O	GSH conjugation, cleavage of Glu, water addition	4.3	C_22_H_32_N_2_O_9_S	501.1925	2.4	4.8
**30**	Hac-CysGly-H_2_O	GSH conjugation, cleavage of Glu; water addition	4.6	C_22_H_32_N_2_O_9_S	501.1917	1.6	3.2
**31**	Hac-CysGly-H_2_O	GSH conjugation, cleavage of Glu; water addition	4.8	C_22_H_32_N_2_O_9_S	501.1881	−2.0	−4.0
**32**	Hac-Cys	Cys conjugation/GSH conjugation, cleavage of Glu and Gly	5.4	C_20_H_27_NO_7_S	426.1599	1.8	4.2
**33**	Hac-H_2_O	water addition	6.0	C_17_H_22_O_6_	323.1480	−1.0	−3.1
**34**	Hac-H_2_O	water addition	6.3	C_17_H_22_O_6_	323.1483	−0.7	−2.2
**35**	Hac-H_2_O-CH_3_	water addition; *O*-methylation	7.9	C_18_H_24_O_6_	337.1657	1.1	3.3
**36**	Hac-OH	hydroxylation	5.4	C_17_H_20_O_6_	321.1314	−1.9	−5.9
**37**	Hac-OH	hydroxylation	6.7	C_17_H_20_O_6_	321.1332	−0.1	−0.3
**38**	Hac-H_2_O-OH	water addition, hydroxylation	5.8	C_17_H_22_O_7_	339.1448	1.0	2.9
**39**	DHac	hydrogenation	6.9	C_17_H_22_O_5_	307.1525	−1.5	−4.9

^a^ Mass accurancy of the used QqTOF MS system (Bruker micrOTOF-Q II) is specified as ±2 mDa (*m*/*z* < 400) and ±5 ppm (*m*/*z* > 400) for signals with intensity >10,000 counts). The calculated mass deviations are within these specifications.

## Data Availability

Data reported in this study is contained within the article. The underlying raw data is available on request from the corresponding author. The raw data are not publicly available due to the complexity and amount of data which requires special software for processing.

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
