# Peer review of "In Vitro Metabolism of Helenalin Acetate and 11α,13-Dihydrohelenalin Acetate: Natural Sesquiterpene Lactones from Arnica"

_metabolites, 2022, doi:10.3390/metabo12010088_

Round 1

Reviewer 1 Report

This paper titled “In vitro metabolism of helenalin acetate and 11α,13-dihydro-helenalin acetate, natural sesquiterpene lactones from Arnica” is interesting. This manuscript could be considered for publication in Metabolites after revising. 

My comments are as follow:

  1. Figure 2 and 4 is not clear, please support high solution photos.
  2. Check the abbreviations, the text format of full text, especially figures. 
  3. Please compare your data with previous studies in the result and discussion section.

    This paper titled “In vitro metabolism of helenalin acetate and 11α,13-dihydro-helenalin acetate, natural sesquiterpene lactones from Arnica” is interesting. This manuscript could be considered for publication in Metabolites after revising. 

    My comments are as follow:

    1. Figure 2 and 4 is not clear, please support high solution photos.
    2. Check the abbreviations, the text format of full text, especially figures. 
    3. Please compare your data with previous studies in the result and discussion section.

Author Response

Reviewer 1 (Responses in red)

This paper titled “In vitro metabolism of helenalin acetate and 11α,13-dihydro-helenalin acetate, natural sesquiterpene lactones from Arnica” is interesting. This manuscript could be considered for publication in Metabolites after revising. 

My comments are as follow:

  1. Figure 2 and 4 is not clear, please support high solution photos.
  • We have revised Figures 2 and 4. Character size has been increased and the figures should now be clearly legible.
  1. Check the abbreviations, the text format of full text, especially figures. 
  • We added the abbreviation of the European Pharmacopoeia (Ph. Eur.) in Line 52.
  • In the Conclusion section, the reintroduction of the abbreviations Hac and DHac was removed (Line 479).
  • Regarding the text format of full text, we used the metabolites template and only changed the width of Figures 2, 4 and 6 to full page width to improve readability.
  1. Please compare your data with previous studies in the result and discussion section.
  • We added a further comparison of our data with three more previous studies (citations [36, 37, 38]) in the result and discussion section (Lines 369-384) in addition to those included previously (e.g. [33] and [34]).

We thank the reviewer for the time and effort spent to help us improve our manuscript!

Reviewer 2 Report

The opinion of this reviewer is that the manuscript is well thought out and the amount of experiments conducted with UHPLC / MS is impressive.

However, the results presented, as well as the study of the experiments, seem to diverge from the outcomes predicted by the authors.

In detail, the differences found are as follows:

  • Authors should have taken into consideration all the phase I metabolic reactions (Hydration, Hydrolysis, Reduction and Oxidation) that can occur on an STL, due to the presence of ketone groups and unsaturations favoring tautomerism.
  • Authors only considered glucuronidation and sulfation, neglecting to consider the reactions of acetylation and methylation.
  • The combination of Phase I and Phase II reactions doesn't describe the physiological competition of molecules for these reactions, because they usually are consequent, take place in different cellular compartments and with different kynetics, so generally are not as competitive as they described.
  • As described in several sections of the manuscript, Arnica tinctures are made from alcoholic extracts. However in the methods the authors listed: "Hac and DHac were isolated from (385 g) dried and pow-355 dered Arnica montana flowerheads by Soxhlet extraction with dichloromethane." This means they used a solvent capable to select the interested and investigated molecules, but it doesn’t prove that the same molecules are highly present in alcoholic tinctures.
    In fact, authors reported that STLs are the main constituents of Arnica tinctures, it would be appropriate to report the sources of this information, citing scientific manuscripts in which Arnica ethanol extracts have been characterized.
    So, at the moment authors provide no evidence about the presence and concentration of Hac and DHac in the alcoholic extracts.
  • I would have preferred a previous characterization of the Arnica extract, even by simple and cheap colorimetric spectrophotometric assays.

Author Response

Reviewer 2 (Responses in red)

The opinion of this reviewer is that the manuscript is well thought out and the amount of experiments conducted with UHPLC / MS is impressive.

However, the results presented, as well as the study of the experiments, seem to diverge from the outcomes predicted by the authors.

In detail, the differences found are as follows:

Authors should have taken into consideration all the phase I metabolic reactions (Hydration, Hydrolysis, Reduction and Oxidation) that can occur on an STL, due to the presence of ketone groups and unsaturations favoring tautomerism.

  • We added the expected phase I metabolites due to structural features in the introduction (Lines 61-64). In the performed experiments all of the mentioned phase I metabolic reactions were taken into consideration and were reported in the Results and Discussion section.

Authors only considered glucuronidation and sulfation, neglecting to consider the reactions of acetylation and methylation.

  • We investigated the major phase II enzymes that are involved in the metabolism of clinically used drugs (UGTs, SULTs, GSTs) [17]. To clarify this point, we added information in the introduction (Lines 69-72) and wrote a comment in the results section 2.2 (Lines 153-157).
  • Although we did not add reagents for methylation several O-methylated-conjugates were formed (maybe due to residual amounts of SAM and methyltransferases in the microsomes mix).

The combination of Phase I and Phase II reactions doesn't describe the physiological competition of molecules for these reactions, because they usually are consequent, take place in different cellular compartments and with different kinetics, so generally are not as competitive as they described.

  • We describe the phase I and phase II reactions as competitive because of the high reactivity of the STLs with GSH which is ubiquitous in the mammalian tissues. This conjugation (phase II reaction) blocks reactive sites of the molecule that would otherwise be hydroxylated or hydrated (phase I reactions) as seen in the phase I experiment.
  • We have added a short statement that this experiment mimics, but doesn’t fully represent, physiological conditions (Lines 174-175).

As described in several sections of the manuscript, Arnica tinctures are made from alcoholic extracts. However in the methods the authors listed: "Hac and DHac were isolated from (385 g) dried and powdered Arnica montana flowerheads by Soxhlet extraction with dichloromethane." This means they used a solvent capable to select the interested and investigated molecules, but it doesn’t prove that the same molecules are highly present in alcoholic tinctures.

  • For pharmaceutical use of Arnica STLs, an ethanolic extract is used. For isolation and analysis of STLs in the laboratory, they are usually also extracted with other organic solvents, mostly dichloromethane. To show the common use of dichloromethane in the extraction of STLs, we added one sentence in the introduction and cited three articles from other working groups in which dichloromethane was used as extraction agent (Lines 59-60).

In fact, authors reported that STLs are the main constituents of Arnica tinctures, it would be appropriate to report the sources of this information, citing scientific manuscripts in which Arnica ethanol extracts have been characterized. So, at the moment authors provide no evidence about the presence and concentration of Hac and DHac in the alcoholic extracts.

  • We did not report, that STLs are the main constituents, but we reported that STLs are the main active substances, which is well established in Arnica research. We made this sentence more concrete to avoid misunderstandings (Line 55). In addition, we added two citations of scientific manuscripts [11,12] that give STL levels in Arnica flowers and the ethanolic Arnica tincture, respectively and report on the anti-inflammatory activity of the STLs:
    • [11] Perry, N.B.; Burgess, E.J.; Rodríguez, G., M. A.; Romero Franco, R.; López Mosquera, E.; Smallfield, B.M.; Joyce, N.I.; Littlejohn, R.P. Sesquiterpene lactones in Arnica montana: helenalin and dihydrohelenalin chemotypes in Spain. Planta Med 2009, 75, 660-666, doi:10.1055/s-0029-1185362.
    • [12] Schmidt, T.J.; Matthiesen, U.; Willuhn, G. On the stability of sesquiterpene lactones in the officinal Arnica tincture of the German pharmacopoeia. Planta Med 2000, 66, 678-681, doi:10.1055/s-2000-8635.

I would have preferred a previous characterization of the Arnica extract, even by simple and cheap colorimetric spectrophotometric assays.

  • Arnica extracts are already well characterized in scientific articles, e.g.:
  • Kriplani, P.; K., G.; S., B.U. Arnica montana L. – a plant of healing: review. Journal of Pharmacy and Pharmacology 2017, 69, 925-945, doi:10.1111/jphp.12724.
  • Kimel, K.; Godlewska, S.; Krauze-Baranowska, M.; Poblocka-Olech, L. HPLC-DAD-ESI/MS ANALYSIS OF ARNICA TM CONSTITUENTS. Acta Poloniae Pharmaceutica ñ Drug Research 2019, 76, 1015-1027, doi:10.32383/appdr/112187.

In addition, the STLs in an Arnica tincture product was characterized by us in:

  • Robledo, S.M.; Vélez, I.D.; Schmidt, T.J. Arnica Tincture Cures Cutaneous Leishmaniasis in Golden Hamsters. Molecules 2018, 23, doi:10.3390/molecules23010150.

Therefore, in the present study, a renewed characterization was not necessary so that we could focus on the metabolites of the sesquiterpene lactones.

We thank the reviewer for the time and effort spent to help us improve our manuscript!

Reviewer 3 Report

The manuscript In vitro metabolism of helenalin acetate and 11α,13-dihydro- helenalin acetate, natural sesquiterpene lactones from Arnica, by Jürgen et al., describes the in vitro metabolism of two sesquiterpene lactones present in Arnica. The study was conducted using rat liver microsomes, pig liver microsomes and human liver microsomes.

The manuscript is well written and interesting for the readers of Metabolites.

The authors have performed an exhaustive and complete analysis of the metabolites and have identified them.

- I consider that authors can enrich the discussion section with some of these papers:

Ziwei Yu , Ke Yang , Ziqiang Chen , Qijuan Li , Zecheng Huang , Wenjun Wang , Siyu Zhao & Huiling Hu (2020): What dominates the changeable pharmacokinetics of natural sesquiterpene lactones and diterpene lactones: a review focusing on absorption and metabolism, Drug Metabolism Reviews, DOI: 10.1080/03602532.2020.1853151

Peng Z, Wang Y, Gu X, Guo X, Yan C. Study on the pharmacokinetics and metabolism of costunolide and dehydrocostus lactone in rats by HPLC-UV and UPLC-Q-TOF/MS. Biomed Chromatogr. 2014 Oct;28(10):1325-34. doi: 10.1002/bmc.3167.

Mengyue Wang, Renjie Xu, Ying Peng, Xiaobo Li, "Metabolism Analysis of Alantolactone and Isoalantolactone in Rats by Oral Administration", Journal of Chemistry, vol. 2018. Article ID 2026357, 7 pages, 2018. https://doi.org/10.1155/2018/2026357

Zhou B, Ye J, Yang N, Chen L, Zhuo Z, Mao L, Liu Q, Lan G, Ning J, Ge G, Yang L, Shen Y, Wang S, Zhang W. Metabolism and pharmacokinetics of alantolactone and isoalantolactone in rats: Thiol conjugation as a potential metabolic pathway. J Chromatogr B Analyt Technol Biomed Life Sci. 2018 Jan 1;1072:370-378. doi: 10.1016/j.jchromb.2017.11.039.

- Page 2, line 83: reference “Fan, 2013” should be numbered.

Author Response

Reviewer 3 (Responses in red)

The manuscript In vitro metabolism of helenalin acetate and 11α,13-dihydro- helenalin acetate, natural sesquiterpene lactones from Arnica, by Jürgen et al., describes the in vitro metabolism of two sesquiterpene lactones present in Arnica. The study was conducted using rat liver microsomes, pig liver microsomes and human liver microsomes.

The manuscript is well written and interesting for the readers of Metabolites.

The authors have performed an exhaustive and complete analysis of the metabolites and have identified them.

- I consider that authors can enrich the discussion section with some of these papers:

Ziwei Yu , Ke Yang , Ziqiang Chen , Qijuan Li , Zecheng Huang , Wenjun Wang , Siyu Zhao & Huiling Hu (2020): What dominates the changeable pharmacokinetics of natural sesquiterpene lactones and diterpene lactones: a review focusing on absorption and metabolism, Drug Metabolism Reviews, DOI: 10.1080/03602532.2020.1853151

Peng Z, Wang Y, Gu X, Guo X, Yan C. Study on the pharmacokinetics and metabolism of costunolide and dehydrocostus lactone in rats by HPLC-UV and UPLC-Q-TOF/MS. Biomed Chromatogr. 2014 Oct;28(10):1325-34. doi: 10.1002/bmc.3167 .

Mengyue Wang, Renjie Xu, Ying Peng, Xiaobo Li, "Metabolism Analysis of Alantolactone and Isoalantolactone in Rats by Oral Administration", Journal of Chemistry, vol. 2018. Article ID 2026357, 7 pages, 2018. https://doi.org/10.1155/2018/2026357

Zhou B, Ye J, Yang N, Chen L, Zhuo Z, Mao L, Liu Q, Lan G, Ning J, Ge G, Yang L, Shen Y, Wang S, Zhang W. Metabolism and pharmacokinetics of alantolactone and isoalantolactone in rats: Thiol conjugation as a potential metabolic pathway. J Chromatogr B Analyt Technol Biomed Life Sci. 2018 Jan 1;1072:370-378. doi: 10.1016/j.jchromb.2017.11.039.

  • Thank you for the suggested articles that fit well with our research. We compared our results with three of them in the results and discussion section (Lines 369-384).

- Page 2, line 83: reference “Fan, 2013” should be numbered.

  • We corrected the literature reference (now in Line 93).

We thank the reviewer for the time and effort spent to help us improve our manuscript!

Reviewer 4 Report

In this work, the Authors present for the first time the results of phase I and phase II in vitro metabolism experiments (as well as a combination of both) of two representative Arnica sesquiterpene lactones. Their studies showed that the main metabolic pathway of the studied compounds was found to be GSH conjugation followed by subsequent reactions (mainly degradation reactions). Moreover, various phase I metabolites of the studied sesquiterpene lactones were detected and identified.

The aim of the work was fully achieved. The studies are well-conducted, -organized, and clearly described. The Materials and Methods section provides enough details. The manuscript and data are sufficiently sound to support the conclusions. In general, it is an interesting work and in my opinion, the manuscript is suitable for publication as it is.

Two minor changes are suggested:

  • Line 61: I think that it should be: γGlu-Cys-Gly as GSH = γ-L-glutamyl-L-cysteinyl-glycine.
  • Line 83: Please provide the correct reference number here instead of [Fan 2013].

Author Response

Reviewer 4 (Responses in red)

In this work, the Authors present for the first time the results of phase I and phase II in vitro metabolism experiments (as well as a combination of both) of two representative Arnica sesquiterpene lactones. Their studies showed that the main metabolic pathway of the studied compounds was found to be GSH conjugation followed by subsequent reactions (mainly degradation reactions). Moreover, various phase I metabolites of the studied sesquiterpene lactones were detected and identified.

The aim of the work was fully achieved. The studies are well-conducted, -organized, and clearly described. The Materials and Methods section provides enough details. The manuscript and data are sufficiently sound to support the conclusions. In general, it is an interesting work and in my opinion, the manuscript is suitable for publication as it is.

Two minor changes are suggested:

Line 61: I think that it should be: γGlu-Cys-Gly as GSH = γ-L-glutamyl-L-cysteinyl-glycine.

  • We corrected the GSH notation (now in Line 67).

Line 83: Please provide the correct reference number here instead of [Fan 2013].

  • We corrected the literature reference (now in Line 93).

We thank the reviewer for the time and effort spent to help us improve our manuscript!

Round 2

Reviewer 1 Report

The author addressed all my comments. 

Reviewer 2 Report

Although some suggestions have been endorsed by the authors, the manuscript is now of significant scientific interest, clarified in the main points and relevant to the scientific community.